# Comparability of ActivPAL-Based Estimates of Meeting Physical Activity Guidelines for Preschool Children

**DOI:** 10.3390/ijerph16245146

**Published:** 2019-12-16

**Authors:** Wendy Yajun Huang, Eun-Young Lee

**Affiliations:** 1Department of Sport and Physical Education, Hong Kong Baptist University, Hong Kong, China; 2School of Kinesiology and Health Studies, Queen’s University, Kingston, ON K7L 3N6, Canada; eunyoung.lee@queensu.ca

**Keywords:** activPAL, young children, physical activity

## Abstract

The activPAL (PAL Technologies, Glasgow, UK) has been increasingly used on children to assess sedentary time and physical activity (PA). However, there is no consensus on how it can estimate PA at different intensities. This study compared three commonly used, activPAL-based classifications of moderate-to-vigorous physical activity (MVPA) (daily steps, acceleration counts, and step rate) in determining compliance with the World Health Organization (WHO)’s PA guidelines for preschool children on a daily basis. One hundred and fourteen preschool children aged 3–6 years wore an activPAL^TM^ for 24 h over 7 consecutive days and provided valid data for a total of 548 days. MVPA was calculated based on published cut-points of counts (MVPA-counts) and step rate (MVPA-step rate). Compliance with standard PA guidelines (≥180 min/day of PA including ≥60 min/day of MVPA) was determined based on three criteria: ≥11,500 steps/day, a threshold of 1418 acceleration counts/15 s, and 25 steps/15 s for MVPA. Applying cut-points of daily steps and acceleration counts provided the same estimates of compliance with the WHO PA guidelines (20%), while the estimated compliance based on the step rate was lower (7.7%). There was a moderate agreement between the daily steps- (or counts-) derived and step rate-derived compliances (κ = 0.41; 95% confidence interval (CI): 0.31, 0.51). The amount of MVPA derived from counts (1.95 ± 0.72 h/day) was significantly higher than that from step rates (0.47 ± 0.31 h/day). The activPAL may be useful for surveillance studies to estimate total PA in preschool children. Further development of the activPAL algorithms based on either counts or step rate is warranted before it can be used to accurately estimate MVPA in this age group.

## 1. Introduction

The World Health Organization (WHO) has recently published guidelines on a range of movement and non-movement behaviors over a 24 h period, which consist of physical activity (PA), sedentary time, and sleep, for children under 5 years [1]. Better monitoring of these behaviors requires objective and accurate measurement in a time-stamped manner, preferably using one single tool to minimize the burden to participants. The thigh-mounted accelerometer activPAL (PAL Technologies, Glasgow, UK) is one such device that has been increasingly used among people of various ages [2,3,4,5]. One of the strengths of activPAL compared to other objective PA measures is that it can be easily made waterproof and enable a 24 h continuous wear protocol [5,6]. Previous studies have supported the use of the activPAL as a reliable and valid measure of sedentary time and PA for children [7,8] and adults [9]. Nevertheless, challenges exist using the activPAL among young children as there is no consensus on the accuracy of estimating PA at different intensities [10]. Ellis et al. assessed PA levels in childcare services using the activPAL and found that 10% of preschoolers aged 3.0–5.9 years achieved the Institute of Medicine (IOM) recommendations for PA (i.e., ≥15 min/h) [11]; however, time spent in moderate-to-vigorous PA (MVPA) was not analyzed in that study, since it is not specified in the IOM recommendations. Given the growing popularity of using the activPAL as a single measurement of both PA and sedentary behavior in preschool-aged children, it is timely to review and compare the available methods to classify MVPA using this device.

The outputs generated from the activPAL, including step rate (cadence), acceleration counts, and daily steps, have been examined for their validity in PA measurement for children and youth [7,12]. Accordingly, three activPAL-based estimates have been utilized to determine time spent in MVPA or approximate sufficient PA (i.e., at least 60 min of MVPA daily). First, the built-in software provides a proprietary algorithm for indirectly estimating metabolic equivalents (METs) via the step rate. Stepping at 120 steps per minute is regarded as 4 MET while the METs for other cadences are calculated based on a formula [13]. Accordingly, stepping at 74 steps per minute is indicative of PA at a moderate intensity (3 MET). However, this algorithm has been consistently found to underestimate METs for light intensity PA (LPA) and MVPA among preschool children [10], 5- to 12-year-olds [12], and adolescent girls [14]. Nonetheless, it showed good classification accuracy of MVPA (defined as ≥3 METs) among children [12] and adults [15]. Rather than using the embedded step rate-based formula, one calibration study was conducted to determine a suitable step rate for approximating moderate intensity, which suggested a threshold of 100 steps per minute as a sensitive and specific estimate of MVPA in adolescent girls [16]. To the best of our knowledge, this is the only available step rate based cut off for MVPA using the activPAL in the literature.

Second, similar to other accelerometers, the activPAL also provides outputs of acceleration counts. One study that included female adolescents aged 15–25 years found that the accelerometer counts had a stronger relationship with indirect calorimetry-assessed METs than the steps did [14]. The authors then argued that a better estimation of MVPA from the activPAL could be achieved by using accelerometry counts rather than the in-built step rate [14]. So far, only two validation studies have examined the activPAL’s counts threshold for classifying MVPA: one for preschoolers [10] and the other for adolescent girls [3]. From these studies, thresholds of 1418 counts/15 s and 2977 counts/15 s were identified as cut-points for MVPA in 4- to 6-year-olds [10] and adolescent girls [3], respectively.

Third, although the suggestions of the current PA guidelines are based on the intensity and duration of physical activities, researchers have attempted to translate these guidelines into step counts. Step-based PA promotion may be more acceptable and understandable to the general public. For preschool children, a daily volume of 10,000 to 14,000 steps was suggested as approximating 60–100 min of MVPA [17]. De Craemer et al. suggested that 11,500 steps a day indicated sufficient PA in preschool children [18]. Vale et al. suggested that a cut-point of 9000 steps a day may be considered as sufficient PA [19]. Both of these studies derived their step count targets based on the accelerometer output, but Craemer et al. applied Reilly’s cut-point (>275 counts/15 s for PA) [20], while Vale et al. applied a cut-point of 200 counts/15 s to determine PA.

Despite the above-mentioned development of step rate and counts-based classifications using the activPAL, few studies have compared their outcomes simultaneously. The only study of this kind was recently published for a small group of adults using one-day data [21]. It was found that the step rate method underestimated—while the counts-based method overestimated—MVPA compared with the ActiGraph [21]. For young children, however, it remains unclear how comparable the different approaches are to estimate MVPA. Therefore, this study compared three commonly used activPAL-based classifications (daily steps, acceleration counts, and step rate) in determining compliance with the WHO PA recommendation on a daily basis.

## 2. Materials and Methods

### 2.1. Participants and Procedure

Kindergarten education in Hong Kong provides a three-year program for the majority of children aged 3 to 6 years. Accordingly, the local health authority recommends the same physical activity guidelines for children aged up to 6 [22]. Half of the kindergartens operate both half day (HD) and whole day (WD) classes in the same premises. The school duration is 3 to 3.5 h a day for HD and 7 to 7.5 h a day for WD. Participants were recruited from four kindergartens (one WD, one HD, and two providing both HD and WD services). Invitations were sent to approximately 600 parents of children in grades K1 to K3. Among them, 155 agreed for their child to participate in the study (25.4% response rate) and eventually 149 children participated in the data collection. The children who refrained from PA participation because of any diseases or developmental problems were excluded. Parental written consent and children’s assent were obtained. The study was approved by the University’s Committee on the Use of Human and Animal Subjects in Teaching and Research (Ref. No.: 02160127).

During a school visit, the trained research staff attached an activPAL device and distributed a “take-home package” to each child. The activPAL devices were made waterproof by wrapping them in a nitrile sleeve and a 3M Tegaderm transparent dressing (with a cartoon sticker to increase the attractiveness) [5]. The device was attached to the midline of the front thigh of the children. The take-home package included the following: (1) an information sheet with a photo of the device and detailed instructions for parents on how to fit and remove the device, (2) a log diary for parents to record the time and reasons for detaching the device, (3) additional transparent dressings to be used in case the device fell off during the week, and (4) a questionnaire to be completed by the parents. The children were instructed to wear the device for seven consecutive days as much as possible, including when sleeping and showering. 

### 2.2. activPAL Measures and Data Reduction

Two models of the activPAL (activPAL3 micro and activPAL3 vt, PAL Technologies Ltd, Glasgow, UK) were used in this study. Data on limb position were sampled at 10 Hz, and this information was used to estimate the time spent sitting/lying, standing upright, or stepping in 15 s epochs. The data were downloaded using the activPAL software (v7.2.38) (PAL Technologies Ltd, Glasgow, UK). The 15 s Excel files generated by PALanalysis (v8.10.32) (PAL Technologies Ltd, Glasgow, UK) were used to calculate the daily time spent in sitting/lying, standing, and stepping, and the step counts. The first non-sedentary epoch after 7:00 a.m. was identified as rise time [23], since no participants woke before 7:00 a.m. based on manual screening of the data. The last non-sedentary epoch followed by an uninterrupted sedentary periods of more than 2 h was identified as bedtime [23]. The period between rise time and bedtime was calculated as waking hours. Non-wear time was defined as a period with ≥ 60 min of consecutive unbroken zero accelerometry counts during waking time [23]. Consistent with previous studies, any days with non-wear time longer than 240 min were considered invalid [2,5,23]. All time spent stepping was calculated as total PA (TPA). MVPA minutes were estimated by applying two complementary thresholds: acceleration counts ≥1418 counts/15 s [10] and step rate ≥25 steps/15 s [16]. The step rate threshold was not developed specifically for preschool, but since there were no other calibrations using the activPAL, it was applied in the current study. Accordingly, any individual day was classified as either meeting or not meeting the WHO recommended amounts of PA in three ways: (1) TPA ≥ 180 min/day and counts-derived MVPA ≥ 60 min/day; (2) TPA ≥ 180 min/day and step rate-derived MVPA ≥ 60 min/day; and (3) total daily steps ≥11,500 steps/day [18]. A threshold of 11,500 steps/day, rather than 9000 steps/day, was applied in this study because the former threshold was in line with the step count target proposed by other researchers [17]. The outcomes were not taken on average over a week for two reasons: first, the WHO guidelines recommend sufficient PA participation every day; second, the pre-determined criteria for minimal valid days will inevitably affect the mean values and reduce the available sample size during the data reduction process.

### 2.3. Anthropometric and Other Variables

Trained staff measured the body weight and height of the participants. The body mass index (BMI) was calculated and the body weight status was classified as either non-overweight or overweight (including obesity) based on the international criteria [24]. The responding parents reported their own age, sex, and highest educational attainment, as well as the age and sex of their children. Educational attainment was classified based on categories in Hong Kong, which have been reported elsewhere [25].

### 2.4. Data Analysis

Analyses were performed using IBM SPSS for Windows, version 25.0 (IBM Corp. Armonk, New York, USA). All valid days were included in the data analysis for comparing the three methods, since the PA recommendation is on a daily basis. The significance level was set at 0.05. Daily MVPA minutes derived from counts and step rate were compared using paired *t*-tests. Pearson correlations were performed among counts- and step rate-derived daily MVPA, and total daily steps. Agreement across the three methods to classify a day as meeting or not meeting the WHO’s PA guidelines was compared using Cohen’s kappa (κ) coefficients (0–0.20 as slight, 0.21–0.40 as fair, 0.41–0.60 as moderate, 0.61–0.80 as substantial, and 0.81–1.00 as almost perfect agreement). The data analyses were also stratified by sex, body weight status, and grade.

## 3. Results

### 3.1. Characteristics of the Participants

Of the 149 children with an average age of 4.9 years (over 95% aged under 6 years), 120 provided valid activPAL data for at least one day. Data downloads from six of the activPAL monitors was unsuccessful. Consequently, a total of 548 valid days from 114 children (70 boys and 44 girls) was retained as the analytical data. On average, the children in the analytical sample wore the monitor for 4.8 days, with 82.5% of them providing three or more valid days (Table 1). Over 70% of the respondent parents were mothers. The excluded children were not different from the analytical sample except for having a slightly higher BMI (16.0 ± 1.84 vs. 15.3 ± 0.05, *p* < 0.05). 

### 3.2. TPA, Daily Steps, and MVPA Derived from Counts and Step Rate

Table 2 shows the TPA, daily steps, and MVPA estimates derived from counts and step rates. On average, the children spent 2.33 h of TPA at different intensities and accumulated 10,691 steps a day. The amount of MVPA derived from counts (1.95 ± 0.72 h/day) was significantly higher than that from step rates (0.47 ± 0.31 h/day) for the analytical sample. This difference remained when the participants were stratified by sex and body weight status. A greater difference between the counts- and step rate-derived MVPA estimates was observed for the K1 children than for the other two (older) groups. All the PA-related outcomes did not vary by sex and body weight status. However, children in K2 and K3 accumulated less counts-derived MVPA than those in K1.

As shown in Table 3, the four PA-related outcomes were positively correlated. The correlation coefficients among TPA, daily steps, and counts-derived MVPA were consistently high (all >0.9). Step rate-derived MVPA had a higher association with daily steps (r = 0.869) than with TPA (r = 0.733) and counts-derived MVPA (r = 0.726).

### 3.3. Agreement across the Three Methods (Daily Steps, MVPA Derived from Counts, and MVPA Derived from Step Rate) in Terms of Meeting the WHO PA Guidelines

As shown in Figure 1A, the proportion of days meeting the PA guidelines was exactly the same no matter whether it was estimated from the cut-points of daily steps or MVPA counts (20%). However, the proportion fell to only 7.7% when using the step rate-derived MVPA estimate. This difference was more profound for the younger children in K1 compared with those in K2 and K3. Applying a threshold of 1418 counts/15 s, children had at least 60 min/day of MVPA for over 90% of the days (Figure 1B). The proportion dropped dramatically to 43% and 7.7% when the cut-points of the daily steps and step rate were applied, respectively. Moderate agreement was found between the daily steps- (or counts-) derived and step rate-derived compliances (κ = 0.41; 95% CI: 0.31, 0.51) (shown in Table 4). When stratified by grade, the kappa coefficient was fair for the K1 children (κ = 0.23; 95% CI: 0.06, 0.41) and moderate for the K2 (κ = 0.42; 95% CI: 0.25, 0.60) and K3 children (κ = 0.51; 95% CI: 0.36, 0.67).

## 4. Discussion

This study compared three commonly used activPAL-based classifications of PA, i.e., daily steps, acceleration counts, and step rate, in preschool children aged 3 to 6 years. Applying cut-points of daily steps and acceleration counts provided the same estimates of compliance with the WHO PA guidelines on a daily basis, while the estimated compliance based on the step rate was lower. Estimated MVPA minutes were significantly higher using the count cut-point than the step rate cut-point.

Objective measures of PA in young children should be able to accurately quantify activities at all intensities specified in the guidelines [1]. One interesting finding of this study was that the published cut-points for daily steps [18] and the activPAL acceleration counts [10] in preschool children provided consistent estimates in compliance with the PA guidelines. This may reflect the fact that the step count threshold of 11,500 steps/day used in this study was based on comparisons of four accelerometry-count cut-points for PA [18], including those of Evenson et al. (>25 counts/15 s) [26], Pate et al. (>37 counts/15 s) [27], Reilly et al. (>275 counts/15 s) [20], and Van Cauwenberghe et al. (>372 counts/15 s) [28]. Different accelerometer cut-points inevitably affect the target daily steps: if higher cut-points are applied, more steps are needed to comply with the PA guidelines. For example, applying Pate’s cut-points [29] or a cut-point of ≥200 counts/15 s [19] for at least LPA, step targets of 6000 or 9000 steps per day were identified, respectively, for preschool children. As there is no consensus on accelerometer cut-points and step targets in young children, a provisional step target of 11,500 steps/day (based on Reilly et al.’s cut-points) was proposed after considering the available recommendations [17,18]. It is notable that the currently available daily step recommendations were not developed specifically based on the activPAL counts. No studies have compared count outputs between activPAL and ActiGraph accelerometry, so it is therefore unclear if any differences exist in acceleration counts between these two devices.

Despite the above-mentioned consistency in terms of compliance measurement, using the cut-point of activPAL counts yielded a significantly higher estimate of the number of days during which children had at least 60 min of MVPA than using the daily steps cut-point. In fact, applying the activPAL counts threshold of ≥1418 counts/15 s, which is the only criterion available for classifying MVPA at ≥3 METs in preschool children in the literature [10], the children in this study accumulated an average of two hours of MVPA per day. In other words, over 80% of the activPAL-detected stepping time was classified as MVPA. Notably, this cut-point for young children was significantly lower than that identified among adolescent girls (≥2997 counts/15 s) [3]. This difference is presumed to be due to the age-related physiological and biomechanical differences between preschool children and adolescents, such as resting metabolic rate [10]. Nevertheless, the appropriateness of using this activPAL MVPA cut-point in young children under free-living conditions remains unknown because the validation was conducted in a room calorimeter [10]. The optimistic estimate of counts-based MVPA needs to be interpreted with caution given the widely reported low compliance with PA guidelines in young children, both locally [30] and worldwide [31,32]. Notably, that validation study used estimates of, rather than measured, basal metabolic rate (BMR) [10]. A study examining the energy cost of PA in preschool children found differences in the predicted METs of selected activities depending on whether estimated or directly measured BMR was used [33]. In addition, the threshold of ≥1418 counts/15 s was calibrated using three METs to classify MVPA. It has been suggested that a higher value of four METs as an alternative indicator of moderate intensity PA for 6- to 17-year-olds [34]. Using a higher MET value for defining MVPA, one may expect a higher threshold for the counts cut-point, and therefore fewer MVPA minutes recorded.

Applying the activPAL step rate cut-point of 25 steps/15 s developed by Harrington et al. [16] provides the most conservative estimates of PA compared with the other two methods; on this basis, 7.7% of the days were classified as having sufficient PA. Among the concerns of using cadence-based estimates is the assumption of continuous walking throughout the test period, which may not accurately reflect real-life walking conditions [35] especially for high step rates per-minute (i.e., a higher intensity of PA) [15]. Contrary to the counts-based estimates, only one fifth of the recorded stepping time was classified as MVPA in this case. The Harrington cut-point is adolescent-specific [16]; therefore, its appropriateness for use in young children remains unclear. So far, no validation of the activPAL step rate has been conducted in preschool-aged and school-aged children. A threshold of 25 steps/15 s (i.e., 100 steps per minute) is consistent with the generally accepted heuristic threshold of moderate intensity activity (≥3 METs) in adults [36], however, it is higher than the activPAL’s built-in METs algorithm for classifying MVPA using a 3-MET threshold (equivalent to 74 steps per minute). In other words, using a cut-point of 100 steps per minute seems to underestimate time spent in MVPA. Interestingly, researchers found that the activPAL’s METs algorithm exhibited a good classification accuracy for MVPA compared with other criteria in 5–12-year-old children [12], and a very high degree of accuracy (Intraclass correlation coefficient = 0.99) in adults in a free-living setting [15]. In this study, disagreement between the step rate and acceleration count in classifying MVPA was more profound among the younger age group in the current study. A uniform step rate cut-point, therefore, may not be appropriate, because higher step rates are required to indicate moderate intensity for younger age groups [34]. Specifically, Rose-Jacobs’s study showed that 3-year-olds exhibited more steps per minute than 5-year-olds at various speeds (from 1.9 km/h to 4.6 km/h) [37].

This study did not attempt to compare the accuracy of the three activPAL-based PA classifications, as no criterion measure of energy expenditure was implemented. Nevertheless, it provided a robust comparison among available methods and a discussion of their potential impact on the PA outcomes. Although it is difficult to conclusively decide the best method to determine the compliance with the PA guidelines in preschool children, using cut-points of daily steps and acceleration counts yielded comparable results and provided similar estimates as those reported in other countries [31,32]. However, the currently available recommendation of 10,000 to 14,000 steps per day for preschool children aged 4–6 years is based on limited evidence [17]. Future step-count recommendations should be better informed by health outcomes of interest. 

Point estimates of MVPA warrant particular attention. More importantly, further development of counts- and step rate-based activPAL algorithms is required to enhance the accuracy of MVPA assessment. For adolescents, stronger associations have been observed between activPAL counts and measured METs than those between step rate and measured METs [14]. Whether this also holds among preschool children needs to be identified. A recent study compared agreement between activPAL and ActiGraph (ActiGraph, LLC, Pensacola, FL, USA) in measuring MVPA for adults [21]. It was found that using a threshold of step rate ≥ 100 steps/min underestimated, while using a cut-point of 1418 counts/15 s overestimated, time spent in MVPA compared to the ActiGraph outputs [21]. Future validation of counts thresholds in young children should also consider using four METs as the indicator of moderate intensity, to be conducted under free-living conditions. The step rate-based activPAL algorithm for estimating MVPA needs significant improvement. Given the consistent underestimation of PA among various age groups [10,12,14], the current step rate cut-point of 100 steps per minute is not recommended for use in preschool children. Furthermore, using the epoch of 15 s may not be able to capture the sporadic and spontaneous nature of preschoolers’ movements. However, previous validation studies of the accelerometer were usually conducted under laboratory or controlled environments, and the findings did not support the notion that short epochs (e.g., 5 s and 10 s) performed better than longer epochs (e.g., 30 s and 60 s) for preschool children [38,39]. Nevertheless, a shorter epoch may be more appropriate to characterize free-living PA for this age group [39,40]. The limitation of using two models of activPAL in this study is acknowledged. Although both the activPAL3 micro and activPAL3 vt are the second generation of the monitors with the same aP3 standard (http://www.palt.com/pals/), there have been no published studies comparing the outcomes from these two models directly.

## 5. Conclusions

This study demonstrated considerable disagreement between methods based on the activPAL counts and step rate for estimating and classifying MVPA in preschool children. However, the estimation of compliance with the PA guidelines is comparable using step count targets and the count threshold. In order to improve the accuracy of the point estimates of MVPA, further development of the activPAL algorithms, based on either counts or step rate is warranted in preschool children. Researchers who wish to use a single device to assess various movement/non-movement behaviors should note these methodological uncertainties when interpreting their findings. The activPAL may be useful for surveillance studies to assess overall PA, though estimates of compliance with PA guidelines will depend on how MVPA were determined.

## Figures and Tables

**Figure 1 ijerph-16-05146-f001:**
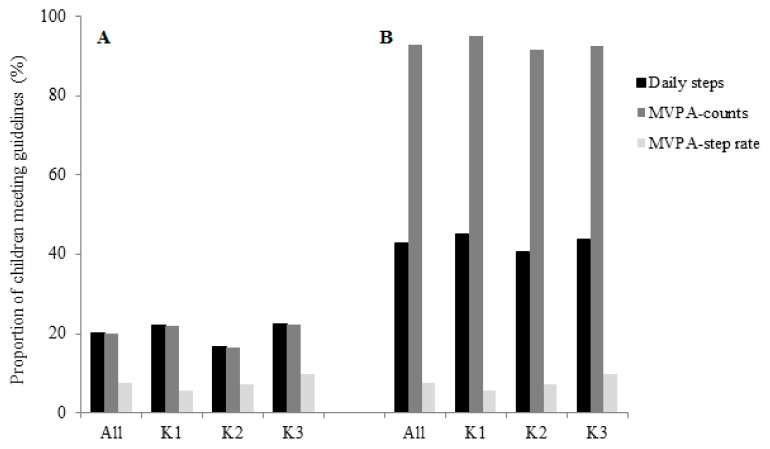
Proportion of days (**A**) meeting WHO PA guidelines, i.e., ≥180 min/day of TPA including ≥60 min/day of MVPA or daily steps ≥11,500 steps/day, and (**B**) having ≥60 min/day of MVPA, according to acceleration counts ≥1418 counts/15 s; step rate ≥25 steps/15 s. MVPA: moderate-to-vigorous physical activity; TPA: total physical activity; WHO, World Health Organization.

**Table 1 ijerph-16-05146-t001:** Characteristics of the whole sample and analytical sample.

Variables	Whole Sample (*n* = 149)	Analytical Sample (*n* = 114)
Sex (% boys)	60.4	61.4
Age (years)	4.9 (0.8)	4.9 (0.8)
Body mass index, BMI (kg⋅m^2^)	15.5 (1.7)	15.3 (1.5)
Number of valid wear days	3.8 (2.6)	4.8 (2.0)
Sex of responding parent (% mother)	74.8	70.8
Parental age (years)	36.9 (7.0)	36.7 (7.2)
Parental education (%)		
Lower secondary or less	27.3	25.0
Completed secondary	46.5	47.9
Tertiary	27.2	27.1

**Table 2 ijerph-16-05146-t002:** TPA, daily steps, and MVPA derived from counts and step rate.

Variables	TPA (Hour/Day)	Daily Steps (Steps/Day)	MVPA-Counts (Hour/Day)	MVPA-Step Rate (Hour/Day)
All	2.33 (0.75)	10,691 (3894)	1.95 (0.72)	0.47 (0.31)
Sex				
Boys	2.35 (0.79)	10,754 (4123)	1.98 (0.77)	0.47 (0.33)
Girls	2.29 (0.69)	10,597 (3526)	1.90 (0.65)	0.48 (0.27)
Grade				
K1	2.38 (0.74)	10,739 (3795)	2.14 (0.72)	0.44 (0.28)
K2	2.28 (0.75)	10,565 (3828)	1.86 (0.71) *	0.48 (0.29)
K3	2.34 (0.76)	10,794 (4051)	1.90 (0.72) *	0.50 (0.34)
Body weight status				
Non-overweight	2.33 (0.75)	10,713 (3913)	1.95 (0.73)	0.47 (0.30)
Overweight	2.19 (0.75)	10,350 (3743)	1.86 (0.69)	0.52 (0.37)

Counts- and step rate-derived MVPA estimates are significantly different for the whole analytical sample and all of the stratified groups. * *p* < 0.05 compared with K1. MVPA: moderate-to-vigorous physical activity; TPA: total physical activity.

**Table 3 ijerph-16-05146-t003:** TPA, daily steps, and MVPA derived from counts and step rate.

Variables	TPA (Hour/Day)	Daily Steps (Steps/Day)	MVPA-Counts (Hour/Day)	MVPA-Step Rate (Hour/Day)
TPA (hour/day)	-	0.957	0.948	0.733
Daily steps (steps/day)	-	-	0.920	0.869
MVPA-counts (hour/day)	-	-	-	0.726

MVPA: moderate-to-vigorous physical activity; TPA: total physical activity.

**Table 4 ijerph-16-05146-t004:** Agreement across the three methods in terms of meeting PA guidelines (kappa coefficients and 95% CI).

Variables	Daily Steps vs. MVPA-Counts	Daily Steps vs. MVPA-Step Rate	MVPA-Counts vs. MVPA-Step Rate
All	1.00	0.41 (0.31, 0.51)	0.41 (0.31, 0.51)
Grade			
K1	1.00	0.23 (0.06, 0.41)	0.23 (0.06, 0.41)
K2	1.00	0.42 (0.25, 0.60)	0.42 (0.25, 0.60)
K3	1.00	0.51 (0.36, 0.67)	0.51 (0.36, 0.67)

CI: confidence interval; MVPA: moderate-to-vigorous physical activity; PA: physical activity.

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
