# Peer review of "Comparability of ActivPAL-Based Estimates of Meeting Physical Activity Guidelines for Preschool Children"

_ijerph, 2019, doi:10.3390/ijerph16245146_

Round 1
Reviewer 1 Report
The manuscript provides comparison on the differences of MVPA derived from step counts and step rate. The discussion clearly demonstrated the disagreement between methods on the ActivPAL-based classification of MVPA in preschool children.
The authors conclude that “Estimation of compliance with the PA guidelines is heavily dependent on data processing decisions.” (line267-268). This statement seems to be obvious. It will benefit readers more if authors can make a recommendation on how to determine the ActivPAL counts and step rate for estimating MVPA in preschool children based on the findings or from the available literature.
Author Response
Thank you for the comment. We agree with the reviewer that this sentence may not be specific. We have now incorporated the following recommendation to our conclusions: “This study demonstrated considerable disagreement between methods based on the activPAL counts and step rate for estimating and classifying MVPA in preschool children. However, estimation of compliance with the PA guidelines is comparable using step count targets and the count threshold. In order to improve the accuracy of point estimates of MVPA, further development of the activPAL algorithms based on either counts or step rate is warranted in preschool children. Researchers who wish to use a single device to assess various movement/non-movement behaviors should note these methodological uncertainties when interpreting their findings. The activPAL may be useful for surveillance studies to estimate the compliance with PA guidelines, but not for the purposes of assessing MVPA for this age group due to its limited ability in distinguishing varying PA intensities.” (lines 311-321)
Reviewer 2 Report
This manuscript aims to compare preschool children's compliance with international guidelines when using different criteria applied to activPAL data. This study finds that compliance estimates differ depending on the criteria used.
Main comments
1) One of the main limitations of this study is how compliance with guidelines has been defined. Guidelines specify engaging in TPA and/or MVPA on a daily basis, but it is unclear whether the every day (all valid days meet specified guidelines) or average day method (e.g. average of 60 min of MVPA across valid days) has been used to determine compliance. This is a critical point of difference which greatly influences estimates. The authors should clearly identify the method used and justify why this has been used.
2) Little information has been provided about how the activPAL data were reduced for analysis. For example, it is highly likely that sleep data were collected given the 24 hour wear protocol, but the authors do not explain how these data were removed from the analyses. Did the 20 hour wear time inclusion criteria include sleep? Is this inclusion criteria appropriate for this age group? How was waking wear determined? How were the final outcomes obtained (e.g. was software, a macro, etc used to reduce data)?
3) Two models of activPAL were used to collect data. Are the collected data comparable? Does the sampling rate affect the data collected?
4) The rationale for the study does not really align with the aims of the study. The rationale discusses a lot of issues relating to determining data decisions to apply to the data, as opposed to why it is important to establish whether different approaches give different estimates of compliance with guidelines. This should be addressed in the manuscript. In addition, the authors do not justify why these have selected the approaches to be used (e.g. 11,500 steps/day vs 9000 steps/day) in the methods. Please provide this in the manuscript.
5) The discussion is rather confusing to follow. The authors attempt to show that compliance varies due to data reduction decision methods, but issues such as 3 vs 4 METs to define MVPA, errors in detecting step counts using the activPAL, using a 15 sec epoch to assess activity etc have not really been noted. For example, research shows children accumulate activity in short bouts, which when using a 15 sec epoch could be missed if it does not exceed the threshold used. On the flip side, the accumulation of steps ignores the accumulation metrics, which might explain some of the differences. The authors are encouraged to reflect on what this might mean for the field moving forwards,
6) It would have been interesting to see more discussion of the implications of these findings. For example, can we be certain that children are (or are not) meeting guidelines? What does this mean for programming/interventions?
Specific comments
Lines 2-3: The title does not reflect the study. Please revise.
Line 31: This sample includes 6 year olds. Does this mean different recommendations should be examined for this age group?
Line 35: Why is it important to determine postures?
Line 44-45: True, but this study is not using a gold standard for assessing MVPA.
Line 57-58, line 67-68: This study does not appear to do this. As such, this is arguably redundant.
Line 78: There is a lack of standardisation using different methods for the same outcome. I wonder if this study should have addressed this, rather than compare different methods?
Line 106: Was there any reason that event files were not used?
Line 128: Including children with at least one valid day could be problematic when assessing compliance. Is this data reduction decision valid? Did the number of valid days impact on compliance estimated.
Line 142: Was this difference significant?
Table 2: It is interesting that for MVPA-counts and MVPA step rate that as MVPA-counts estimates reduce across age and weight status, MVPA-step rate estimates increases. Why do the authors think these findings go in opposite directions? This could be discussed in the discussion section.
Table 3: The TPA column and MVPA-step rate row do not contain data. Suggest removing.
Line 165-174: Please clarify whether the text is referring to section A or B of the figure for clarity.
Table 4: There is perfect agreement between daily steps and MVPA-counts. Why? What are the implications of this?
Line 190: Don't the results show that how compliance is determined influenced the findings?
Line 197-201: All of the cut-points mentioned here relate to ActiGraph acceleration cut-points. Has anyone looked at activPAL cut-points and step guidelines? Does this matter?
Line 222: It isn't clear why 3 METs was used over 4 METs. Please clarify.
Round 2
Reviewer 2 Report
Thank you to the authors for taking the time to consider the comments raised and provide additional information into some of the decisions made. There are a couple of minor points that I feel could be addressed to further enhance the manuscript, and these are outlined below.
1) Response to comment 1: I am still a little confused over how compliance with guidelines has been defined. In the initial comment, I wanted clarification over whether a child was classed as being compliant with guidelines based on all of the valid measurement days exceeding 60 mins of MVPA, or whether an average of all valid days exceeded 60 mins. For example, if a child wore the monitor for 3 days and achieved 59, 60, and 61 minutes of MVPA, they would not be compliant with guidelines based on definition 1, but would be compliant using definition 2. This is the information that is needed in the manuscript, as it is critical to understanding the results. Please can this be added/revised for clarity.
2) Response to comment 3: I think it is prudent to acknowledge this a potential limitation in this study.
3) Response to comment 6: I don't really agree with the final sentence in the conclusion. Isn't compliance based on the ability of the monitor to detect MVPA? It appears that the implications of this study are that the activPAL can be used within surveillance studies, though depending on how MVPA was determined impacted on estimates.
4) Response to comment 8: This was really useful information. Please consider including some of these points in the manuscript.
Author Response
Thank you very much for your positive feedback and comments. Please see the attachment.
